# CLIP-FREE, LABEL-FREE, ZERO-SHOT CONCEPT BOTTLENECK MODELS

## ABSTRACT

Concept Bottleneck Models (CBMs) map dense, high-dimensional feature representations into a set of human-interpretable concepts which are then combined linearly to make a prediction. However, modern CBMs rely on the CLIP model to establish a mapping from dense feature representations to textual concepts, and it remains unclear how to design CBMs for models other than CLIP. Methods that do not use CLIP instead require manual, labor intensive annotation to associate feature representations with concepts. Furthermore, all CBMs necessitate training a linear classifier to map the extracted concepts to class labels. In this work, we lift all three limitations simultaneously by proposing a method that converts any frozen visual classifier into a CBM without requiring image-concept labels (label-free), without relying on the CLIP model (CLIP-free), and by deriving the linear classifier in a zero-shot manner. Our method is formulated by aligning the original classifier's distribution (over discrete class indices) with its corresponding vision-language counterpart distribution derived from textual class names, while preserving the classifier's performance. The approach requires no ground-truth image–class annotations, and is highly data-efficient and preserves the classifiers reasoning process. Applied and tested on over 40 visual classifiers, our resulting CLIP-free, zero-shot CBM sets a new state of the art, surpassing even supervised CLIP-based CBMs. Finally, we also show that our method can be used for zero-shot image captioning, outperforming existing methods based on CLIP, and achieving state-of-art results.

## 1 INTRODUCTION

Visual classifiers predict a class as a linear combination of dense, high-dimensional visual feature vectors that are difficult to interpret by humans. Concept Bottleneck Models (CBMs) (Koh et al., 2020) address this challenge by mapping these feature vectors into a set of human-interpretable concepts, each associated with an activation score (referred to as a concept activation). Predictions are then made as a linear combination of these concept activations. Initial CBMs required image–concept annotations to train the bottleneck layer that maps dense features to concepts. Modern CBMs (Oikarinen et al., 2023; Yang et al., 2022; Panousis et al., 2023; Rao et al., 2024) overcome this limitation by leveraging the CLIP model (Radford et al., 2021) as an annotation-free alternative. Since CLIP models map image and text into a shared embedding space, these approaches can query image features against a set of predefined textual concepts within that space and use cosine similarity scores to find a matching annotation. These approaches are commonly referred to as *label-free CBMs*.

Nevertheless, in many real-world scenarios a high-performing, task-specific *legacy* model already exists and typically outperforms zero-shot CLIP models (Zhang et al., 2024). A natural question that then arises is: *how to develop a label-free CBM for such legacy specialist models?* Retraining such a specialist on a large image–text corpus following the CLIP approach is impractical, in terms of computational cost and need of a huge amount of image-text data. Furthermore, obtaining ground-truth image-concept annotations is time consuming and labor intensive. Finally, retraining this legacy model further alters its original decision-making process and distribution, which is typically not desired. As a result, CBMs remain confined to CLIP-like models, leaving specialist vision models aside with no clear way to adapt them into the CBM framework.

In this work, we lift this limitation by first proposing a method dubbed as TextUnlock, which aligns a distribution of a frozen visual classifier to its corresponding vision-language counterpart, without

relying on CLIP. TextUnlock has four important properties: First, it is *efficient*; it is inexpensive to train and can be performed on any standard hardware, regardless of the size of the original classifier. The number of data points is also significantly reduced compared to CLIP-based approaches. Secondly, it is *label-free*, no labels are required to achieve this formulation. Thirdly, TextUnlock is *trained to preserve* the original distribution and reasoning process of the classifier, and does not compromise the classifier's original performance (average of 0.2 points drop in accuracy). Finally, our method is applicable to *any* vision architecture, whether convolutional-based, transformer-based or hybrid. After the original classifier's distribution is aligned to its vision-language counterpart distribution with TextUnlock, we simply query the transformed classifier's image features against a set of predefined text concepts to obtain concept activations for our CBM, and then derive the concept-to-class classifier directly from the transformed classifier's text source, without requiring any additional training but rather operating in a zero-shot manner (i.e., we do not train a linear probe to associate the concept activations to class labels).

In summary, our contributions are as follows: **(i)** We propose a method to convert any frozen visual classifier into a CBM without relying on the CLIP bottleneck (CLIP-free), neither annotated image-concept data (label-free) while also deriving the concept-to-class classifier in a zero-shot manner. **(ii)** We demonstrate the effectiveness of our method with 40 different architectures, along with extensive ablation studies and intervention results, and further show how our method can also be used to perform zero-shot image captioning. **(iii)** Our method sets new state-of-the-art results, outperforming existing works including supervised CLIP-based CBMs, despite being trained only on ImageNet-1K.

## 2 RELATED WORK

**Concept Bottleneck Models (CBMs):** Concept Bottleneck Models (CBMs) (Koh et al., 2020) are a class of inherently interpretable models that map dense representations into human-understandable concepts, and then make predictions based on a linear combination of these interpretable units rather than opaque features. Recently, Label-Free CBMs (LF-CBMs) have been introduced, which leverage CLIP to perform the feature-to-concept mapping without requiring annotated image–concept data. This can be achieved either by using CLIP to provide ground-truth image–concept similarity scores that train the bottleneck layer (Oikarinen et al., 2023), or by querying image features against a predefined set of concepts in the CLIP space (Yang et al., 2022; Panousis et al., 2023) and using the resulting cosine similarities as concept activations. Several follow-up works (Rao et al., 2024; Knab et al., 2024; Benou & Riklin-Raviv, 2025; Yuksekgonul et al., 2023) adopt this CLIP-based pipeline. However, these approaches remain limited to CLIP-based classifiers and cannot be applied to other visual models. Moreover, they require training a linear probe to map concept activations to class predictions. Our work addresses both limitations: we propose a fully CLIP-free CBM and demonstrate that our method can be readily used to derive the linear probe in a fully zero-shot manner.

**Transforming Visual Features to Text:** There exist works that try to decode visual features of models beyond CLIP. DeVIL (Dani et al., 2023) and LIMBER (Merullo et al., 2023) train an autoregressive text generator to map visual features into image captions, leveraging annotated image-caption pairs as ground-truth data. Notably, all these works (1) rely on annotated datasets and (2) explicitly train the generated text to align with what annotators want the visual features to describe, thereby altering the classifier's original reasoning process. ZS-A2T (Salewski et al., 2023) converts attention maps into natural language in a zero-shot manner using Large Language Models (LLMs), but is constrained to vision-language models trained to learn a shared vision-language embedding space (e.g., through a contrastive objective). Text-to-Concept (T2C) (Moayeri et al., 2023) also trains a linear layer to map image features of any classifier into the CLIP vision encoder space, such that they can be interpreted via text using the CLIP text encoder. Notably, all these methods rely on the CLIP approach and/or its supervision, and they alter the classifier's distribution by entirely discarding its output class distribution. In contrast, our method is CLIP-free, can be applied to any pretrained classifier without requiring any annotated data, and explicitly preserves the classifier's predictive distribution.

## 3 METHOD

We first elaborate on our proposed TextUnlock method in Section 3.1, which will then allow us to design zero-shot concept bottleneck models, which we discuss in Section 3.2.

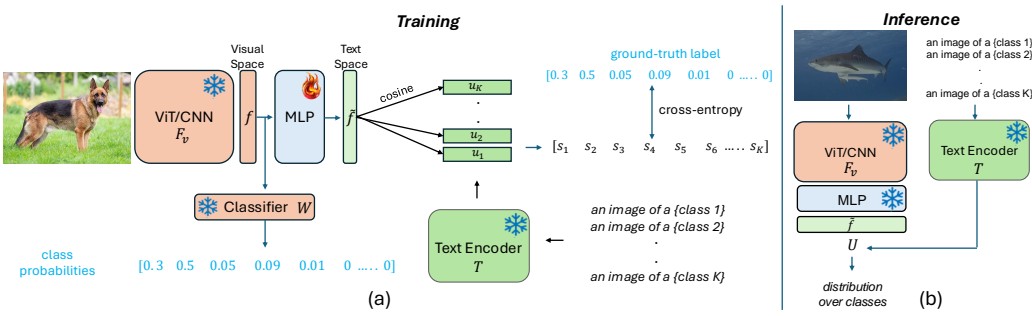

Figure 1: **Overview of our proposed TextUnlock. (a)** The process of training the MLP mapping between vision and text space with pseudocode given in Appendix Section A. **(b)** The process of inference with the adapted visual classifier. The text encoder acts as weight generator for a linear classifier. ❄ indicates that the module is frozen, while 🔥 indicates trainable.

## 3.1 TEXTUNLOCK

A visual classifier assigns an image to a specific category from a predefined set of discrete class labels. For example, in ImageNet-1K (Deng et al., 2009), this set contains $1,000$ class labels. Originally, these discrete class labels correspond to class names in text format. For example, in ImageNet-trained models, the discrete label $1$ corresponds to the class *goldfish*. These classes are typically discretized to facilitate training with cross-entropy. However, when the textual class names are used, they provide an advantage. Specifically, when the textual class names are embedded into vector representations (e.g., using a word embedding model or text encoder), they provide semantic information. Specifically, These embeddings reside in a continuous space where nearby vectors capture related semantic associations. For the "goldfish" example, neighboring vectors might include terms such as "freshwater", "fins" and "orange". Such associations can be viewed as high-level conceptual attributes that characterize the class. Our method learns to map images into this text embedding space using *only* class names, thus linking both the class name *and its surrounding semantic associations* with the image. This process thus naturally supports unseen words that are not part of the class names (e.g., concepts in the CBM). We accomplish this through a trainable multilayer perceptron (MLP) that projects the visual features into the text embedding space, and is explicitly trained to match its distribution with the original classifier's class distribution. This is done while keeping both the visual and textual encoders frozen. By using solely the class names without any supplementary information, we can learn a semantically meaningful image-text space. This allows us to query the visual classifier with text queries beyond the class names (e.g., concepts) and obtain concept activations for CBMs.

Consider an image $I$ and a visual classifier $F$ composed of a visual feature extractor $F_v$ and a linear classifier $W$. Note that $F$ can be of any architecture. $F_v$ embeds $I$ into an $n-$dimensional feature vector $f \in \mathbb{R}^n$. That is, $f = F_v(I)$. The linear classifier $W \in \mathbb{R}^{n \times K}$ takes $f$ as input and outputs a probability distribution $o$ for the image across $K$ classes. That is, $o = \text{softmax}(f.W) \in \mathbb{R}^K$. For ImageNet-1K, $K = 1000$. Consider also any off-the-shelf text embedding model $T$ which takes in an input text $l$ and embeds it into a $m-$dimensional vector representation $u \in \mathbb{R}^m$. That is, $u = T(l)$. Note that $u$ and $f$ are not in the same space and can have a different number of dimensions, so we cannot query $f$ with the text $l$.

We propose to learn a lightweight MLP mapping function that projects the visual features $f$ into the text embedding space of $T$, resulting in a new vector $\tilde{f}$. That is, $\tilde{f} = \text{MLP}(f)$, where $\tilde{f} \in \mathbb{R}^m$. Note that the visual encoder $F_v$, the linear classifier $W$, and the text encoder $T$ are all frozen; only the MLP is trainable, making our method *parameter-efficient*. We then take the textual class names of the $K$ classes, and convert each into a text prompt $l^p$, represented as: "an image of a {class}" where {class} represents the class name in text format. This results in $K$ textual prompts, each of which is encoded with $T$: $u_i = T(l_i^p), \forall i = 1, \ldots, K$. Stacking all the encoded prompts, we get a matrix $U \in \mathbb{R}^{K \times m}$. Here, $U$ acts as weights of the classification layer for our approach. We then calculate the cosine similarity[1] between each $u_i$ and the visual features $f$: $s_i = \tilde{f}.u_i$. Equivalently, this can be

---

[1]in the rest of this paper, we will omit the unit norm in cosine similarity to reduce clutter, and represent it with the dot product.

performed as a single vector-matrix multiplication: $S = \tilde{f}.U^T$, where $S \in \mathbb{R}^K$ represents the cosine similarity scores between the visual features and every text prompt $l_i^p$ representing a class. In other words, $S$ represents the classification logits of our approach.

The most straightforward way to training the MLP is to leverage the ground-truth labels from the dataset, aligning $S$ with the ground-truth distribution. However, this approach violates two key desiderata: (1) it necessitates annotated data, and (2) re-training the legacy classifier alters its original decision distribution $o$, thereby changing the reasoning process of the classifier (i.e., how it maps visual features to class probabilities and makes predictions). Notably, the original soft probability distribution $o$ is a function of the linear classifier $W$, so $W$ cannot be ignored. We instead propose to align $S$ to the original decision distribution $o$ through cross-entropy loss. For a single sample, the loss is given by

$$L = -\sum_{i=1}^{K} o_i \log \left( \frac{e^{s_i}}{\sum_{j=1}^{K} e^{s_j}} \right) \ . \tag{1}$$

This task can be viewed as a knowledge distillation problem, except that we do not distill the knowledge of a bigger teacher model to a smaller student model, but instead **distill the distribution of the original model to its counterpart vision-language distribution**. Note that the loss is equivalent to the KL divergence loss between $o$ and the predicted distribution, since the additional entropy term $H(o)$ that appears in the KL divergence loss is a constant that does not depend on the MLP parameters. Eq. 1 shows that our approach is label-free and steers the MLP to be faithful to the original classifier $F$, since it is explicitly trained for that purpose. Note also that this loss function naturally encodes the classifier's relationship across all classes.

We provide an illustration of the TexUnlock approach in Figure 1 and a PyTorch-like pseudocode in Section A of the Appendix. It is important to note that we only use the class name to formulate the textual prompt $l^p$, and no other supplementary information such as class descriptions, concepts or hierarchies (see Section O in the Appendix for more information).

After training, the projected visual features and the text encoder features lie in the same space. We can therefore query the visual features with any text by finding the alignment score between the embedded text and the projected visual features. In the case of image classification, the text queries remain the class prompts, and encoding them with the text encoder $T$ is equivalent to generating the weights of a linear classifier for the classification task formulated as $\text{argmax}(\tilde{f}.U^T)$, see Figure 1(b).

## 3.2 ZERO-SHOT CONCEPT BOTTLENECK MODELS

Once the classifier's distribution is matched to its corresponding vision–language counterpart distribution via TextUnlock, we proceed to formulate the proposed zero-shot CBMs. Note that at this stage, all model components (including the MLP) are frozen and no further training is performed. CBMs consist of two steps: (1) **concept discovery**, followed by (2) **concept-to-class prediction**. In step (1), the dense output features of a visual encoder are first mapped to textual concepts (e.g., words or short descriptions of objects) each with a score that represents the concept activation to the image. In step (2), a linear classifier $W^{con}$ is trained on top of these concept activations to predict the class.

**Concept Discovery.** We remind readers from Section 3 that $U \in \mathbb{R}^{K \times m}$ is the output of the text encoder $T$ for the class prompts, which in essence represent the classification weights of the newly formulated classifier. We assume that we are given a large set of predefined textual concepts, denoted as $\mathcal{Z}$, and with cardinality $|\mathcal{Z}| = Z$. Following other works (Rao et al., 2024), and without loss of generality, we use the $Z = 20K$ most common words in English (Google, 2016) as our concept set. These are general concepts that are sufficiently expressive and represent world knowledge and are not tailored towards any specific dataset. We ablate on other concept sets in Appendix Section I for the interested reader. To ensure that the concepts are meaningful, we apply a rigorous filtering procedure to the concept set. Specifically, we remove any terms that exactly match the target class name, as well as any constituent words that form the class name (for example, eliminating "tiger" and "shark" when the class name is "tiger shark"). In addition, we exclude terms corresponding to the parent and subparent classes (e.g., "fish" and "animal" for the class "tiger shark"), other species within the same category, and any synonyms of the target class name. Details of this procedure can be found in Section N of the Appendix. This systematic filtering guarantees that the resulting concept set is free of terms that are overly similar or directly derived from the target classes. With this

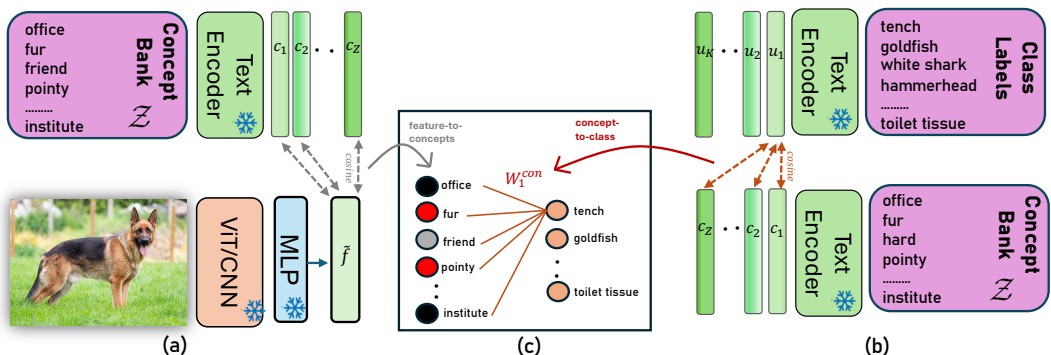

Figure 2: **Building zero-shot CBMs for any pretrained classifier.** We first perform **(a)** concept discovery, followed by **(b)** building the concepts-to-class classifier in a zero-shot manner, which results in **(c)** our final CBM. Note that the concept bank only needs to be encoded once.

filtering procedure, since no concept appears in the training data, our concept discovery set becomes entirely zero-shot. This will also demonstrate the MLP's ability to generalize to the semantic space surrounding the class name.

We use the same text encoder $T$ that generates the linear classifier $U$ to generate concept embeddings, by feeding each concept $z_i \in \mathcal{Z}$ to the text encoder $T$ to generate a concept embedding $c_i$. That is, $c_i = T(z_i), \forall i = 1, \ldots, Z$. By performing this for all $Z$ concepts, we obtain a concept embedding matrix $C \in \mathbb{R}^{Z \times m}$. For an image $I$, we extract its visual features $f$ and use the MLP to map them to $\tilde{f}$ which now lies in the text embedding space. That is, $\tilde{f} = \text{MLP}(f)$, and $\tilde{f} \in \mathbb{R}^m$. Since $C$ and the mapped visual features $\tilde{f}$ are now in the same space, we can query $\tilde{f}$ to find which concepts it responds to. That is, we perform concept discovery using the cosine similarity between $\tilde{f}$ and each row-vector in $C$. The concept activations are obtained by $\tilde{f}.C \in \mathbb{R}^Z$ and represent the activation score for each of the $Z$ concepts. We provide an illustration in Figure 2(a).

**Concept-to-class Prediction.** The classifier $W^{con}$ takes in concept activations and outputs a distribution $S_{cn}$ over classes. We build $W^{con}$ in a zero-shot manner, where here *zero-shot* indicates that no training is required to map concept activations to classes. Recall that both $U$ and $C$ are outputs of the text encoder $T$, and they are already in the same space. Therefore, we can build the weights of the classifier $W^{con}$ with a text-to-text search between the concepts and the class name. Specifically, we calculate the cosine similarity between the concept embeddings $C$ and the classification matrix $U$ to obtain the new weights for $W^{con}$. That is, we perform $C \cdot U^T \in \mathbb{R}^{Z \times K}$. Therefore, the weights of $W^{con}$ represent how similar the class name is to each of the concepts. This process is shown in Figure 2(b). In total, the output distribution $S_{cn}$ of the CBM is defined as:

$$S_{cn} = \underbrace{(\tilde{f} \cdot C^T)}_{\text{concept discovery}} \cdot \underbrace{(C \cdot U^T)}_{\text{concept-to-class}} = \tilde{f} \cdot \underbrace{C^T C}_{\text{gram matrix}} \cdot U^T . \tag{2}$$

From Eq. 2 we make an interesting observation. Our formulation involves scaling the linear feature-based classifier $U$ by the gram matrix of concepts ($C^T C \in \mathbb{R}^{m \times m}$). The gram matrix represents a feature correlation matrix measuring how different dimensions of the feature space relate to each other. Notably, if the gram matrix is the identity ($C^T C = I$), we get back our original feature-based classifier given by $\tilde{f}.U^T$. Therefore, to convert any classifier to a CBM, we plug in the gram matrix in-between, making it a convenient way to directly switch to an inherently interpretable model. Eq. 2 also shows that we do not change the linear classifier $U$, we only scale it by the gram matrix of concepts. This means our CBMs preserve the basic reasoning process of the original classifier. By this, we obtain zero-shot CBMs that discover concepts and build $W^{con}$ in a zero-shot manner for any classifier (Figure 2(c)). An additional unique property is the construction of CBMs *at inference time*, allowing any concept set to be selected to build a CBM on-the-fly. This makes our method highly flexible with respect to the chosen concept set.

| Model | Top-1 | Orig. | $\Delta$ | Model | Top-1 | Orig. | $\Delta$ |
|---|---|---|---|---|---|---|---|
| ResNet50 | 75.80 | 76.13 | $-0.33$ | ResNeXt50-32x4d$_{v2}$ | 80.79 | 80.88 | $-0.09$ |
| ResNet50$_{v2}$ | 80.14 | 80.34 | $-0.20$ | ResNeXt101-64x4d | 83.13 | 83.25 | $-0.12$ |
| ResNet101$_{v2}$ | 81.50 | 81.68 | $-0.18$ | ResNeXt101-32x8d | 79.10 | 79.31 | $-0.21$ |
| ResNet101 | 77.19 | 77.37 | $-0.18$ | ViT-B/16 | 80.70 | 81.07 | $-0.37$ |
| WideResnet50 | 78.35 | 78.47 | $-0.12$ | ViT-B/16$_{v2}$ | 84.94 | 85.30 | $-0.36$ |
| WideResNet50$_{v2}$ | 81.17 | 81.31 | $-0.14$ | ViT-L/32 | 76.72 | 76.97 | $-0.25$ |
| WideResNet101$_{v2}$ | 82.21 | 82.34 | $-0.13$ | ViT-L/16 | 79.56 | 79.66 | $-0.10$ |
| DenseNet161 | 77.04 | 77.14 | $-0.10$ | ViT-L/16$_{v2}$ | 87.61 | 88.06 | $-0.45$ |
| DenseNet169 | 75.46 | 75.60 | $-0.14$ | Swin-Base | 83.22 | 83.58 | $-0.36$ |
| EfficientNetv2-S | 84.04 | 84.23 | $-0.19$ | Swinv2-Base | 83.72 | 84.11 | $-0.39$ |
| EfficientNetv2-M | 84.95 | 85.11 | $-0.16$ | BeiT-B/16 | 84.54 | 85.06 | $-0.52$ |
| ShuffleNetv2$_{x2.0}$ | 75.83 | 76.23 | $-0.40$ | BeiT-L/16 | 87.22 | 87.34 | $-0.12$ |
| ConvNeXt-Small | 83.42 | 83.62 | $-0.20$ | DINOv2-B | 84.40 | 84.22 | $+0.18$ |
| ConvNeXt-Base | 83.88 | 84.06 | $-0.18$ | ConvNeXtV2-B | 84.56 | 84.73 | $-0.17$ |
| ConvNeXt-B$_{pt}$ | 85.27 | 85.52 | $-0.25$ | ConvNeXtV2-B$_{pt}$ | 86.07 | 86.25 | $-0.18$ |
| ResNeXt50-32x4d | 77.44 | 77.62 | $-0.18$ | ConvNeXtV2-B$_{pt}$@384 | 87.34 | 87.50 | $-0.16$ |

Table 1: **TextUnlock-ed visual classifiers maintain classification performance.** Comparison of our re-formulated classifiers for several models. Top-1 indicates our results of the new formulation, and Orig. denotes the original Top-1 accuracy. $\Delta$ represents their difference ($\Delta$ = Top-1 $-$ Orig).

## 4 EXPERIMENTS

We first provide results of the the classifier with its corresponding vision-language distribution aligned using our method TextUnlock. We use the ImageNet-1K benchmark dataset due to the widespread publicly available visual classifiers trained and evaluated on it. In Section F of the Appendix, we also report results on other datasets including Places365 (Zhou et al., 2017), EuroSAT (Helber et al., 2019) and DTD (Cimpoi et al., 2013) showing that our method is also applicable to domain-specific and fine-grained datasets. We apply TextUnlock on a diverse set of 40 visual classifiers. For CNNs, we consider the following family of models (each with several variants): Residual Networks (ResNets) (He et al., 2015), Wide ResNets (Zagoruyko & Komodakis, 2016), ResNeXts (Xie et al., 2016), ShuffleNetv2 (Ma et al., 2018), EfficientNetv2 (Tan & Le, 2021), Densely Connected Networks (DenseNets) (Huang et al., 2016), ConvNeXts (Liu et al., 2022) and ConvNeXtv2 (Woo et al., 2023). For Transformers, we consider the following family of models (each with several variants): Vision Transformers (ViTs) (Dosovitskiy et al., 2021), DINOv2 (Oquab et al., 2024), BeiT (Bao et al., 2022), the hybrid Convolution-Vision Transformer CvT (Wu et al., 2021), Swin Transformer (Liu et al., 2021b) and Swin Transformer v2 (Liu et al., 2021a). All models are pretrained on ImageNet-1K from the PyTorch (maintainers & contributors, 2016) and HuggingFace (Wolf et al., 2020) libraries. Models with the subscript *pt* indicate that the model was pretrained on ImageNet-21k before being finetuned on ImageNet-1K. Models with a subscript *v2* are trained following the updated PyTorch training recipe (Vryniotis, 2021). Finally, BEiT, DINOv2 and ConvNeXtv2 are pretrained in a self-supervised manner before being finetuned on ImageNet-1k. When training with TextUnlock, both the pretrained classifier and text encoder remain frozen, only the MLP is trained on the ImageNet training set following Eq. 1.

Performance is evaluated using the same protocol and dataset splits as the original classifier, specifically the 50,000 validation split of ImageNet-1K. For the text encoder, we use the MiniLM Sentence Encoder (Wang et al., 2020) as it is powerful, fast and efficient. We provide ablation studies on other text encoders in Section D of the Appendix. Results are presented in Table 1. We report the replicated Top-1 accuracy of the re-formulated classifier with our method in the first column, the original Top-1 accuracy of the classifier in the second column, and the difference between them ($\Delta$) in the last column. As it can be seen, the loss in performance as indicated by $\Delta$ is minimal, with an average drop in performance of approximately 0.2 points across all models. We also perform ablation studies on the MLP to verify its design, impact, role and that it learns meaningful transformations in Section C of the Appendix (see Section C.1 for the role and impact of the MLP and Section C.2 for its design hyperparameters). We also evaluate our transformed classifier robustness to prompt

| Supervised CBMs | | | | | |
|---|---|---|---|---|---|
| **Method** | **Model** | **Top-1** | **Method** | **Model** | **Top-1** |
| LF-CBM | CLIP ResNet50 | 67.5 | LF-CBM | CLIP ViT-B/16 | 75.4 |
| LaBo | CLIP ResNet50 | 68.9 | LaBo | CLIP ViT-B/16 | 78.9 |
| CDM | CLIP ResNet50 | 72.2 | CDM | CLIP ViT-B/16 | 79.3 |
| DCLIP | CLIP ResNet50 | 59.6 | DCLIP | CLIP ViT-B/16 | 68.0 |
| DN-CBM | CLIP ResNet50 | 72.9 | DN-CBM | CLIP ViT-B/16 | 79.5 |
| DCBM-SAM2 | CLIP ViT-L/14 | 77.9 | DCBM-RCNN | CLIP ViT-L/14 | 77.8 |
| Zero-Shot CBMs (Ours) | | | | | |
| **Method** | **Model** | **Top-1** | **Method** | **Model** | **Top-1** |
| ZS-CBM | ResNet50 | 73.9 | ZS-CBM | ViT-B/32 | 73.3 |
| ZS-CBM | ResNet50$_{v2}$ | 78.1 | ZS-CBM | ViT-B/16 | 79.3 |
| ZS-CBM | ResNet101 | 75.3 | ZS-CBM | ViT-B/16$_{v2}$ | 83.2 |
| ZS-CBM | ResNet101$_{v2}$ | 79.9 | ZS-CBM | Swin-Base | 82.2 |
| ZS-CBM | WideResNet50 | 76.9 | ZS-CBM | Swinv2-Base | 82.6 |
| ZS-CBM | WideResNet50$_{v2}$ | 79.2 | ZS-CBM | ViT-B/16$_{pt}$ | 81.5 |
| ZS-CBM | WideResNet101$_{v2}$ | 81.0 | ZS-CBM | BeiT-B/16 | 83.0 |
| ZS-CBM | DenseNet121 | 69.9 | ZS-CBM | DINOv2-B | 82.6 |
| ZS-CBM | DenseNet161 | 75.2 | ZS-CBM | ConvNeXt-B$_{pt}$ | 84.0 |
| ZS-CBM | EfficientNetv2-S | 83.0 | ZS-CBM | ConvNeXtV2-B$_{pt}$ | 84.9 |
| ZS-CBM | EfficientNetv2-M | 83.9 | ZS-CBM | BeiT-L/16 | 86.2 |
| ZS-CBM | ConvNeXt-Small | 81.9 | ZS-CBM | ViT-L/16$_{v2}$ | 86.3 |
| ZS-CBM | ConvNeXt-Base | 82.8 | ZS-CBM | ConvNeXtV2-B$_{pt}$@384 | **86.4** |

Table 2: **Our zero-shot CBMs outperform CLIP-based counterparts.** Accuracy of Supervised and Zero-Shot CBMs (ZS-CBMs, ours) on ImageNet validation set. Similar backbones are color-coded. Best within backbone family is underlined, overall best is **bold**.

variations in Section H of the Appendix. For results on other models, we refer to Section L of the Appendix, and for implementation details, to Section J of the Appendix.

**Zero-Shot Concept Bottleneck Models:** We report CBM evaluation results on the ImageNet validation set using the top-1 accuracy in Table 2. In Section G of the Appendix, we report results on additional datasets including Places365, as well as fine-grained and domain-specific datasets such as DTD and EuroSAT. We compare against six SOTA methods: LF-CBMs (Oikarinen et al., 2023), LaBo (Yang et al., 2022), CDM (Panousis et al., 2023), DCLIP (Menon & Vondrick, 2023), DN-CBMs (Rao et al., 2024), and DCBM (Knab et al., 2024), all using the same concept set for fair comparison. All these methods are supervised and use CLIP-based models for computing the concept activations, and many use it as a backbone as well. Our zero-shot CBMs (ZS-CBMs) outperforms all the supervised CBMs, setting a new state-of-the-art performance. Notably, even a simple ResNet-50 classifier trained solely on ImageNet already outperforms the CBM for the significantly more powerful ResNet-50 CLIP model trained on 400M samples (that is, we used $400\times$ less images and $400,000\times$ less text data). The best results are obtained by the ConvNeXtv2 model, which achieves a top-1 accuracy of $86.4$. All models show close to original accuracy, which means we can transform any classifier to be inherently interpretable without notable performance loss.

**Effectivenss of Concept Interventions.** Another common way to evaluate interpretability of CBMs is concept intervention. We report concept intervention results on our ZS-CBMs in Section E of the Appendix. In this way, we show the effectiveness of the concepts, how we can mitigate biases, debug models and fix their reasoning by explicitly intervening in the concepts of the bottleneck layer to control predictions.

In Figure 3, we present qualitative examples of a selection from the top concepts responsible for the prediction, along with their weight importance on the x-axis. The weight importance is calculated by multiplying the concept activation with its corresponding weight to the predicted class. We use various concept sets to demonstrate the flexibility of our method to any desired concept set directly at test time (on-the-fly), as this process simply involves encoding the chosen concept set using the text encoder. All examples use the LF-CBM concept set (Oikarinen et al., 2023). By observing the first example, the image is predicted as a "scorpion" because it is has a lizard-like overall body, it is a venomous, desert animal with claws on its legs, and has large claws on its hand that look like a crab.

Interestingly, in the second example, the top-detected concept is a "lift arm on the side." Although this is not the primary feature defining a dumbbell, it reflects a well-documented bias in the literature (Samek & Müller, 2019) regarding the "dumbbell" class. Because most training images for this class show a dumbbell being lifted by an arm, the classifier not only learns to recognize the dumbbell but also associates it with the hand or arm that lifts it. With our method, we can obtain a textual interpretation of the biases that the original classifier learns. We also provide qualitative examples of global class-wise concepts detected in Section P of the appendix.

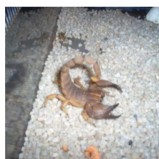 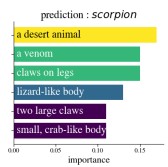 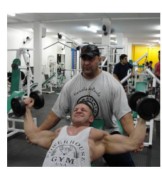 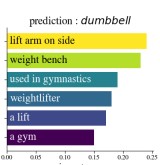 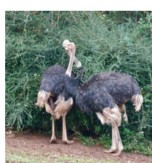 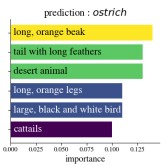

Figure 3: Qualitative examples of our zero-shot CBMs. We show the top-detected concepts, each with their corresponding importance score to the on the x-axis.

## 5 ZERO-SHOT IMAGE CAPTIONING

We also show that TextUnlock enables zero-shot image captioning with any pretrained visual classifier beyond CLIP models. Existing zero-shot captioning methods largely rely on CLIP and its shared vision–language space. Thanks to TextUnlock, zero-shot image captioning can now be performed for any pretrained visual classifier. We adapt the method introduced in ZeroCap (Tewel et al., 2021) for our purpose. Specifically, ZeroCap is a test-time approach that learns to produce a caption that maximizes the similarity with the image features. We first project the visual feature vector $f$ using the MLP to obtain $\tilde{f}$. That is, $\tilde{f} = \text{MLP}(f)$. Since $\tilde{f}$ is now in the same space as the text encoder $T$, we can measure its association to any encoded text. We utilize an off-the-shelf pretrained language decoder model (e.g., GPT-2), denoted as $G$, to generate open-ended text. We keep $G$ frozen to maintain its language generation capabilities and instead use prefix-tuning (Li & Liang, 2021) to guide $G$ to generate a text that maximizes the similarity with the transformed visual feature vector $\tilde{f}$. Specifically, we attach a set of learnable "virtual" tokens to $G$. Denote the generated output text of $G$ for one iteration as $h^j$, where $j$ represents the iteration number. $h^j$ is encoded with the text encoder $T$, to yield a vector $y^j$, That is, $y^j = T(h^j)$. As $y^j$ and $\tilde{f}$ are now in the same embedding space, we maximize the cosine similarity between them in order to update the learnable tokens. We perform this process for several iterations. As this process is not the core contribution of our work, we leave more details to Section M of the appendix, which also includes a detailed illustration in Figure 2.

We now evaluate the performance of the zero-shot captions produced on the COCO image captioning dataset (Lin et al., 2014). Since we do not train any model on the ground-truth image captions provided by COCO, we use zero-shot image captioning as a benchmark. Note that the COCO dataset differs in distribution than ImageNet, as a single image may contain many objects, interactions between them, and categories not included in ImageNet (*e.g.,* person). Therefore, it also serves as a way to evaluate generalization of our method to other datasets, given that we only used the ImageNet images and class names for training. We present results on the widely used "Karpathy test split" with various vision classifiers. As baselines, we compare our approach against existing methods in zero-shot image captioning, specifically ZeroCap (Tewel et al., 2021) and ConZIC (Zeng et al., 2023), both which use CLIP. For evaluation, we employ standard natural language generation metrics: BLEU-4 (B@4) (Papineni et al., 2002), METEOR (M) (Banerjee & Lavie, 2005), ROUGE-L (R-L) (Lin, 2004), CIDEr (C) (Vedantam et al., 2014), and SPICE (S) (Anderson et al., 2016). Results are shown in Table 3. ConvNeXtv2 achieves state-of-the-art performance on CIDEr and SPICE, the two most critical metrics for evaluating image captioning systems. Even with a simple ResNet-50 vision encoder trained on ImageNet-1K (1.2 million images), our approach outperforms the baseline methods on CIDEr and SPICE, despite the latter utilizing the significantly more powerful CLIP vision encoder, trained on 400 million image-text pairs (that is, we used $400\times$ less images and $400{,}000\times$ less text data compared to CLIP). We present qualitative examples of the produced zero-shot image captions from different visual classifers in Figure 4. It is interesting to see how different classifiers "see" the image. From the first example, BeiT-L/16 captures features of both the vegetables and the

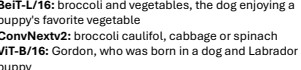
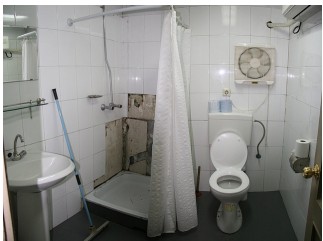
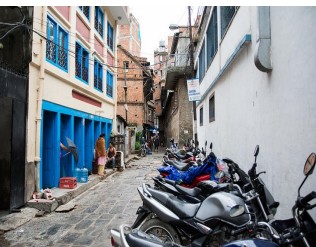

**BeiT-L/16:** broccoli and vegetables, the dog enjoying a puppy's favorite vegetable
**ConvNextv2:** broccoli caulifol, cabbage or spinach
**ViT-B/16:** Gordon, who was born in a dog and Labrador puppy

**BeiT-L/16:** shower curtain, bathing room and toilet.
**ConvNextv2:** shower curtain toilet seat and toilets, which were made by the bath.
**ViT-B/16:** shower curtain in the bathroom, and a small bathtub.

**BeiT-L/16:** motorcycles with the motorized scooters and electric motors
**ConvNextv2:** motorbike, scooter and motorcycle scrapping
**ViT-B/16:** motorcycles and motor scooters, motorcycle engines

Figure 4: Qualitative examples of zero-shot image captioning.

dog, whereas ConvNeXtV2 captures only the vegetables. In contrast, ViT-B/16 focuses exclusively on the dog and its characteristics.

Note that our results in Table 3 are outperformed by the baseline ZeroCap on the BLEU-4 (B4) and METEOR (M) metrics. However, it is important to note that B4 and M are n-gram overlap-based metrics. They assume that the generated caption follows a specific structure and style. We verify this hypothesis by applying compositional image captioning (Kulkarni et al., 2011; Lu et al., 2018) with in-context learning. Specifically, a set of image-grounded verbs and concepts (e.g., attributes, objects) are first detected using the concept discovery step described in Section 3.2, and they are fed along with six in-context examples from the COCO captioning set to a language model, capable of imitating the style and structure of the in-context examples, in order to compose them into a natural sounding sentence. This also demonstrates the direct application of our approach to compositional image captioning. The last row of Table 3 with the superscript *com* shows that the (B4, M and R-L) are boosted, which verifies our hypothesis about the low scores of B4 and M compared to baseline methods. We provide more details and results on other models on this paradigm of compositional image captioning in Section K of the Appendix. We further show how we can use this approach to generate captions tailored to any domain (see Section K.1 in the Appendix).

| Model | B4 | M | R-L | C | S |
|---|---|---|---|---|---|
| ZeroCap | **2.6** | **11.5** | — | 14.6 | 5.5 |
| ConZIC | 1.3 | 11.5 | — | 12.8 | 5.2 |
| **Ours** | | | | | |
| DenseNet161 | 1.50 | 10.2 | 20.4 | 15.8 | 6.3 |
| ResNet50 | 1.43 | 10.2 | 20.3 | 15.9 | 6.2 |
| WideResNet50 | 1.40 | 10.2 | 20.4 | 16.0 | 6.4 |
| WideResNet101$_{v2}$ | 1.50 | 10.4 | 20.5 | 16.6 | 6.4 |
| ResNet101$_{v2}$ | 1.48 | 10.4 | 20.6 | 16.7 | 6.5 |
| ResNet50$_{v2}$ | 1.47 | 10.5 | 20.6 | 16.8 | 6.5 |
| ConvNeXt-B$_{pt}$ | 1.50 | 10.6 | 20.8 | 17.2 | 6.7 |
| DINOv2-Base | 1.50 | 10.7 | 21.0 | 17.3 | 6.7 |
| ViT-B/16$_{v2}$ | 1.50 | 10.5 | 20.9 | 17.3 | 6.5 |
| BeiT-L/16 | 1.50 | 10.6 | 20.9 | 17.6 | 6.9 |
| ViT-B/16$_{pt}$ | 1.50 | 10.7 | 20.9 | 17.7 | 6.9 |
| ConvNeXtV2-B$_{pt}$@384 | 1.60 | 10.7 | **21.1** | **17.9** | **6.9** |
| ConvNeXtV2-B$_{pt}$@384$^{com}$ | **4.40** | **12.7** | **30.2** | **18.7** | **7.2** |

Table 3: Zero-Shot Image Captioning Performance

3 with the superscript *com* shows that the (B4, M and R-L) are boosted, which verifies our hypothesis about the low scores of B4 and M compared to baseline methods. We provide more details and results on other models on this paradigm of compositional image captioning in Section K of the Appendix. We further show how we can use this approach to generate captions tailored to any domain (see Section K.1 in the Appendix).

## 6 CONCLUSION

We introduced a method for transforming any frozen visual classification model into a Concept Bottleneck Model (CBM). We proposed TextUnlock, the core of our method, that aligns the distribution of the original classifier with that of its vision–language counterpart. This method further enables a CLIP-free, label-free conversion of the frozen visual classification model into a CBM, and further allows us to derive zero-shot concept bottleneck concept-to-class classifiers (without training the linear classifier), Applied on 40 different models, our method outperforms supervised CLIP-based CBMs and achieves state-of-the-art results. TextUnlock further allows us to perform zero-shot image captioning on any visual classification model, also achieving state-of-the-art results and surpassing zero-shot captioning baselines based on CLIP. Finally, as with any research work, this study has its own limitations which are discussed in Section B of the Appendix.

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

# A    PSEUDOCODE OF TEXTUNLOCK

We provide a PyTorch-like pseudocode of TextUnlock in Listing 1.

```python
# text_feats: textual features of class names from a frozen sentence encoder, shape (num_classes, text_dim)
# classifier: linear classifier weights of a frozen vision_encoder, shape (visual_dim, num_classes)
# mlp: trainable MLP from visual_dim -> text_dim
# images: batch of B images, shape (N, 3, height, width)

visual_feats = vision_encoder(images) # (N, visual_dim)
logits = visual_feats @ classifier # (N, num_classes)
original_dist = softmax(logits, dim=-1) # (N, num_classes)

mapped_feats = mlp(visual_feats) # (N, text_dim)
mapped_feats = l2_norm(mapped_feats) # (N, text_dim)
text_feats = l2_norm(text_feats) # (N, text_dim)

pred_logits = mapped_feats @ text_feats.T # (N, num_classes)
pred_dist = softmax(pred_logits, dim=-1) # (N, num_classes)

# cross entropy with original model's soft distribution
loss = -(original_dist * log(pred_dist)).sum(dim=1).mean()
loss.backward() # only mapper parameters are updated
```

Listing 1: PyTorch-like pseudocode for TextUnlock

# B    LIMITATIONS

As with any research work, our study has its own limitations that should be transparent and acknowledged. In particular, we identify a primary limitation of our method concerning wrong semantic associations of class names in CBMs. This issue is closely tied to the choice of concept set used. When using the 20K most common words in English as our concept set, we observe that class names get associated to wrong semantically-related concepts. Figure 1 illustrates such cases. In the first example, the bird drake is linked to artist-related concepts (Rihanna, Robbie, lyric). This occurs because the bird "drake" is less familiar to the text encoder than the artist "drake". In fact, a google search with the word "drake" yields directly the artist rather than the bird. In the second example, the top-detected concepts for the prediction "african grey" are incorrect semantic associations with the word "african" (ethiopian, tanzania) and the word "grey" (purple, blue) that do not pertain to the bird itself. We also observe similar cases that may raise ethical concerns (e.g., the animal "cock" leads to associations with male reproductive terms). For each example, we also report the total logit score of the prediction.

However, it is worth noting the following:

1. Meaningful concepts such as "duck" (first example) are still detected among the top concepts.
2. The incorrect semantic associations contribute only a negligible portion of the total logit, accounting for approximately 0.01% of the overall prediction score.
3. This issue is considerably less severe when using alternative concept sets, such as the LF-CBM concept set tailored for ImageNet.
4. This issue also appears in CLIP-based CBMs and hence not unique to our approach.

# C    ABLATION STUDIES OF THE MLP

## C.1    ROLE AND IMPACT OF THE MLP

We first evaluate the role and impact of the MLP to verify its contribution and that it learns meaningful transformations, and does not collapse into a trivial function. We perform the following ablation studies: **1) Mean Ablation:** For an image, we replace the input features to the MLP with a constant mean of the features calculated across the full ImageNet validation set. **2) Random Features:** For an image, we replace the input features to the MLP with random values sampled from a normal distribution with a mean and standard deviation equal to that of the features calculated across the full ImageNet validation set. **3) Random Weight Ablation:** We randomize the weights of the MLP projection. **4) Shuffled Ablation:** For an image, we replace the input features to the MLP with input features of another random image in the validation dataset.

Figure 1: Limitations of our method in wrong semantic concept association

For each ablation experiment, we compute the ImageNet validation accuracy. We expect the accuracy to drop across all ablations. As shown in Table 1, the accuracy nearly drops to zero in every case, confirming the effectiveness of our MLP.

| Model | | Mean Feature ↓ | Random Features ↓ | Shuffled Features ↓ | Random Weights ↓ |
|---|---|---|---|---|---|
| ResNet101v2 | Ours | 81.49 | 81.49 | 81.49 | 81.49 |
| | Ablated | **0.10** | **0.11** | **1.70** | **0.11** |
| ConvNeXt-Base | Ours | 83.88 | 83.88 | 83.88 | 83.88 |
| | Ablated | **0.10** | **0.11** | **1.79** | **0.10** |
| BeiT-L/16 | Ours | 87.22 | 87.22 | 87.22 | 87.22 |
| | Ablated | **0.10** | **0.11** | **1.87** | **0.11** |
| DINOv2-B | Ours | 84.40 | 84.40 | 84.40 | 84.40 |
| | Ablated | **0.10** | **0.13** | **1.76** | **0.09** |

Table 1: Ablation studies of the MLP.

## C.2 MLP DESIGN

Next, we perform an ablation study over the number of layers and output dimension scaling factor (`dim_out_factor`) of the MLP. `dim_out_factor` specifies how much to scale the previous layer's dimensionality. For example, in a ViT-B/16 model the hidden dimension is 768; setting `dim_out_factor = 2` therefore expands the projection from 768 to $2 \times 768 = 1536$. Results are shown in Table 2 for a ResNet50 model. We observe that the 2 layers and a output dimension scaling factor of 2 provides the best results.

| Layers | dim_out_factor | Top-1 (%) |
|---|---|---|
| 1 | 1 | 72.48 |
| 1 | 2 | 74.01 |
| 2 | 1 | 75.41 |
| 2 | 2 | 75.80 |

Table 2: Performance comparison of the MLP for different layer and dimension configurations.

## D ABLATION STUDIES ON THE TEXT ENCODER

In Table 3, we present ablation studies using other text encoders from the Sentence-BERT library (Reimers & Gurevych, 2019) with the ResNet50 visual classifier. We observe that the choice of the

text encoder has minimal effect on the performance. This is because even lower-performing text encoders are capable of understanding class names.

| Text Encoder | Top-1 (%) |
|---|---|
| DistilRoberta | 75.73 |
| MPNet-Base | 75.78 |
| MPNet-Base-MultiQA | 75.76 |
| MiniLM | **75.80** |

Table 3: Ablation studies on other text encoders

## E  EFFECTIVENESS OF CONCEPT INTERVENTIONS

As a supplementary experiment, we also report concept intervention results on our ZS-CBMs. Interventions on CBMs are an effective tool to mitigate biases, debug models and fix their reasoning by explicitly intervening in the concepts of the bottleneck layer to control predictions. The Waterbirds-100 dataset (Sagawa* et al., 2020) is a standard dataset used in previous works (Rao et al., 2024) to conduct CBM intervention experiments. It is a binary classification dataset of two classes: waterbirds and landbirds. The training images of waterbirds are on water backgrounds, and training images of landbirds are on land backgrounds. However, the validation images do not have that correlation, where waterbird images appear on land backgrounds and landbird images on water backgrounds. The model is assumed to learn the water–land background correlation to perform this classification task. By building a CBM, we can correct this bias by intervening in concepts in the CB layer. However, there are two challenges associated with using the Waterbirds dataset in our work, and therefore we curated a validation dataset of waterbirds/landbirds directly from the ImageNet validation set. We provide more details about this process in Section Q. Our curated dataset includes 140 validation images (70 for each class). We create our ZS-CBM using the two class prompts:"an image of a waterbird" (for the waterbird class) and "an image of a landbird" (for the landbird class), and using the same concept set from (Rao et al., 2024) which includes a collection of bird-related concepts and a collection of land-related concepts. The ZS-CBM achieves a low accuracy, as shown in Table 4, which indicates the water-land bias the model performs for classification. To correct this, we intervene in the concepts in the CB layer, following the setup from (Rao et al., 2024). **Intervention R (Int. R):** We zero-out activations of any bird concepts from the bottleneck layer, and expect the accuracy to drop. **Intervention K (Int. K):** We keep activations of bird concepts as they are, but scale down the activations of all remaining concepts by multiplying them with a factor of 0.1, and expect the accuracy to increase. Results are presented in Table 4 for some models, and demonstrate the success of our intervention experiments.

We also conduct concept intervention experiments with a more challenging multi-class setup. Unlike the Waterbirds dataset where bias-correlation issues are assumed, we make no such assumption here. We instead evaluate whether intervening on class-related concepts in the bottleneck layer and zeroing out their activations, impairs model accuracy. A drop in accuracy indicates that these concepts are important for prediction. We select a subset of 10 classes from ImageNet following (Howard, 2019). We use the original ImageNet validation images for those classes, rather than the validation set from (Howard, 2019), because the latter contains original ImageNet training examples that our model has already seen. We employ an LLM to generate 5 highly-relevant concepts for each class, achieving in total 50 concepts. The classes and concepts we used are provided in Section R of the supplementary material. We then measure CBM accuracy before and after intervention. Results are provided in Table 5. For each image, we zero out the activations of its class-related concepts and report the **Intervention R (Int. R)** metric. As shown in Table 5, this intervention reduces accuracy by approximately 20% on average, underscoring the concepts importance to the CBM.

## F  TRANSFORMED CLASSIFIER RESULTS ON OTHER DATASETS

We conduct extra experiments on 3 datasets. Places365 (domain-specific to scenes), DTD (domain-specific to texture and fine-grained), and EuroSAT (domain-specific to satellite images). For each

| Model | Orig. CBM | Int. (R)↓ | Int. (K)↑ |
|---|---|---|---|
| BeiT-B/16 | 54.29 | 41.43 **(-12.86)** | 58.57 **(+4.28)** |
| ConvNeXtV2$_{pt}$@384 | 53.57 | 42.14 **(-11.43)** | 59.29 **(+5.72)** |
| ConvNeXt_B$_{pt}$ | 53.57 | 42.86 **(-10.71)** | 58.57 **(+5.00)** |
| DiNOv2 | 52.86 | 43.57 **(-9.29)** | 59.29 **(+6.43)** |
| BeiT-L/16 | 52.86 | 44.29 **(-8.57)** | 58.57 **(+5.71)** |

Table 4: CBM Interventions on the standard Waterbirds dataset

| Model | Orig. CBM | Int. (R)↓ |
|---|---|---|
| ResNet50 | 96.80 | 76.00 **(-20.8)** |
| ResNet101 | 97.00 | 76.80 **(-20.2)** |
| DINOv2-B | 98.60 | 78.40 **(-20.2)** |
| BeiT-B/16 | 98.60 | 78.80 **(-19.8)** |
| ConvNeXtV2$_{pt}$@384 | 99.40 | 79.40 **(-20.0)** |

Table 5: CBM Interventions in a multi-class setup

dataset, we show the top-1 accuracy of the classifier's original performance and the top-1 accuracy of our transformed classifier. Results are shown in Table 6.

| Dataset | Model | Original (%) | Transformed (%) |
|---|---|---|---|
| Places365 | ResNet50 | 54.77 | 53.90 |
| | DenseNet161 | 56.13 | 55.54 |
| EuroSAT | ResNet50 | 93.95 | 94.23 |
| | WideResNet101 | 93.78 | 93.88 |
| | ViT-B/16 | 93.67 | 93.53 |
| DTD | ResNet50 | 69.47 | 69.26 |
| | WideResNet101 | 68.62 | 68.14 |
| | ViT-B/16 | 69.95 | 69.68 |

Table 6: Results of our transformed classifier on Places365, EuroSAT, and DTD datasets

## G  CBM RESULTS ON OTHER DATASETS

We report CBM results on additional datasets: Places365 (domain-specific to scenes), DTD (domain-specific to texture and fine-grained), and EuroSAT (domain-specific to satellite images). When a baseline method is also reported on that dataset, we include it as a baseline. Otherwise, we use CLIP models as a baseline to act as a feature extractor and to compute concept activations, and train a linear classifier on top of the concept activations, formulating a CLIP supervised baseline. All baselines use the same concept set. Results are shown in Table 7. For Places 365, we can see that the transformed DenseNet161 classifier trained only on ImageNet, outperforms supervised CLIP-based ResNet and ViTs CBM methods in a zero-shot manner. The same applies for EuroSAT and DTD with our transformed Image-Net only trained classifiers. Therefore, the experiments show that our method scales to domain-specific and fine-grained datasets.

## H  PROMPT VARIATIONS

We evaluate the robustness of our transformed classifier to variations in text prompts. Using the transformed ViT-B/16, we provide a diverse set of prompts and measure the Top-1 accuracy on ImageNet. Results are shown in Table 8. We order the prompts from highest to lowest scoring. We

| Dataset | Method | Model | Acc (%) |
|---|---|---|---|
| Places365 | **Supervised** | | |
| | DCLIP | CLIP-ResNet50 | 37.90 |
| | DCLIP | CLIP-ViT-B/16 | 40.30 |
| | LF-CBM | CLIP-ResNet50 | 49.00 |
| | LF-CBM | CLIP-ViT-B/16 | 50.60 |
| | CDM | CLIP-ResNet50 | 52.70 |
| | CDM | CLIP-ViT-B/16 | 52.60 |
| | **Zero-Shot** | | |
| | Ours | ResNet50 | 51.57 |
| | Ours | DenseNet161 | 53.42 |
| EuroSAT | **Supervised** | | |
| | Baseline | CLIP-ResNet50 | 86.27 |
| | Baseline | CLIP-ViT-B/16 | 88.57 |
| | **Zero-Shot** | | |
| | Ours | ResNet50 | 94.22 |
| | Ours | WideResNet101 | 94.12 |
| | Ours | ViT-B/16 | 93.65 |
| DTD | **Supervised** | | |
| | Baseline | CLIP-ResNet50 | 57.77 |
| | Baseline | CLIP-ViT-B/16 | 61.86 |
| | **Zero-Shot** | | |
| | Ours | ResNet50 | 68.88 |
| | Ours | WideResNet101 | 66.97 |
| | Ours | ViT-B/16 | 68.46 |

Table 7: CBM results on Places365, EuroSAT, and DTD datasets

find that our transformed classifier is robust to text prompts. The worst prompt degrades the baseline accuracy by only 0.36 points

| Prompt | Top-1 Accuracy (%) |
|---|---|
| an image of a {} | 80.70 |
| a photo of one {}. | 80.67 |
| a photo of a {}. | 80.66 |
| a close-up photo of a {}. | 80.63 |
| a black and white photo of a {} | 80.63 |
| a close-up photo of the {}. | 80.62 |
| a cropped photo of a {}. | 80.61 |
| a good photo of a {}. | 80.61 |
| a dark photo of a {}. | 80.61 |
| a bright photo of a {}. | 80.60 |
| a bright photo of the {}. | 80.59 |
| a bad photo of a {}. | 80.57 |
| a blurry photo of a {}. | 80.57 |
| a pixelated photo of a {}. | 80.55 |
| a photo of many {}. | 80.54 |
| a low-resolution photo of the {} | 80.54 |
| a low-resolution photo of a {}. | 80.52 |
| a photo of my {}. | 80.45 |
| a jpeg-corrupted photo of a {}. | 80.34 |

Table 8: Effect of different text prompts on Top-1 accuracy using ViT-B/16.

# I    OTHER CONCEPT SETS

Early works such as label-free CBMs use LLMs to automatically generate a set of concepts for a target class. In recent works such as DN-CBM (Rao et al., 2024) and DCBM (Knab et al., 2024), it was however shown that general, pre-defined concept sets outperform the LLM-generated ones, which introduce many spurious correlations and hallucinated associations. In Table 9, we compare general pre-defined and LLM-generated concept sets. Across different types of architectures, general, pre-defined concept sets outperform the LLM-generated ones by several percent. Please note that while earlier works such as LF-CBMs employ LLM-generated concept sets (which underperform compared to the general, pre-defined concept set), all results reported in our main manuscript—including those from prior works—are presented using the general, pre-defined concept set to ensure a fair comparison.

| Model | Predefined | LLM-generated |
|-------|-----------|---------------|
| Swinv2-Base | 82.60 | 80.34 |
| ConvNext-Base | 82.80 | 80.24 |
| ConvNeXtV2-B$_{pt}$@384 | 86.40 | 84.03 |
| ViT-B/16 | 79.30 | 76.37 |

Table 9: Comparison of Predefined and LLM-generated concept sets on CBMs

# J    IMPLEMENTATION DETAILS

For the text encoder, we use the all-MiniLM-L12-v1[2] model available on the Sentence Transformers library (Reimers & Gurevych, 2019). This text encoder was trained on a large and diverse dataset of over 1 billion training text pairs. It contains a dimensionality of $m = 384$ and has a maximum sequence length of 256.

Our MLP projector is composed of 3 layers, the first projects the visual feature dimensions $n$ to $n \times 2$ and is followed by a Layer Normalization (Ba et al., 2016), a GELU activation function (Hendrycks & Gimpel, 2016) and Dropout (Srivastava et al., 2014) with a drop probability of 0.5. The second layer projects the $n \times 2$ dimensions to $n \times 2$ and is followed by a Layer Normalization and a GELU activation function. The final linear layer projects the $n \times 2$ dimensions to $m$ (the dimensions of the text encoder). We train the MLP projector with a batch size of 256 using the ADAM optimizer (Kingma & Ba, 2015) with a learning rate of 1e-4 that decays using a cosine schedule (Loshchilov & Hutter, 2017) over the total number of epochs. We follow the original image sizes that the classifier was trained on.

For the training images, we apply the standard image transformations that all classifiers were trained on which include a Random Resized Crop and a Random Horizontal Flip. For the validation images, we follow exactly the transformations that the classifier was evaluated on, which include resizing the image followed by a Center Crop to the image size that the classifier expects. Each model is trained on a single NVIDIA GeForce RTX 2080 Ti GPU.

# K    COMPOSITIONAL IMAGE CAPTIONING

Compositional Image Captioning is an old image captioning paradigm termed (Kulkarni et al., 2011), later revived with deep learning methods (Lu et al., 2018). In compositional captioning, a set of image-grounded concepts (such as attributes, objects and verbs) are first detected, and a language model is then used to compose them into a natural sounding sentence. With the current advancements of Large Language Models (LLMs) and their powerful capabilities, we use an LLM as a compositioner. Specifically, we detect the top concepts and verbs to the image using the concept discovery method introduced in Section 3.2 and shown in Figure 2(a), and feed them, along with their similarity scores, to an LLM. For the concepts, we use the same concept set as in Section 3.2. We also add a list of the most common verbs (Dat, 2015) in English to the pool. This allows us to cover all possible words and

---

[2]https://huggingface.co/sentence-transformers/all-MiniLM-L12-v1

interactions. We prompt the LLM to utilize the provided information to compose a sentence, given a limited set of in-context examples from the COCO captioning training set (in our experiments, we use 9 examples). This allows us to generate sentences adhering to a specific style and structure. For this experiment, we used GPT4o-mini (OpenAI, 2024) as our LLM, as it is fast and cost-efficient. In the prompt, we explicitly instruct the LLM to refrain from reasoning or generating content based on its own knowledge or assumptions, and that all its outputs must be strictly grounded in the provided concepts, verbs, and score importance. We use the following prompt:

```
I will give you several attributes and verbs that are included in
an image, each with a score.  The score reflects how important
(or how grounded) the attrribute/verb is to the image, and
higher means more important and grounded.  Your job is to
formulate a caption that describes the images by looking at
the attributes/verbs with their associated scores.  You should
not reason or generate anything that is based on your own
knowledge or guess.  Everything you say has to be grounded in
the attrbiutes/verbs and score importances.  Please use the
following the structure, style, and pattern of the following
examples.  Example 1:  A woman wearing a net on her head cutting
a cake.  Example 2:  A child holding a flowered umbrella and
petting a yak.  Example 3:  A young boy standing in front of a
computer keyboard.  Example 4:  a boy wearing headphones using one
computer in a long row of computers.  Example 5:  A kitchen with a
stove, microwave and refrigerator.  Example 6:  A chef carrying
a large pan inside of a kitchen.  Here are the attributes and
scores:  {detected concepts with scores}, and these are the verbs
and scores:  {detected verbs with scores}.
```

Results are shown in Table 10.

| Model | B4 | M | R-L | C | S |
|---|---|---|---|---|---|
| ZeroCap | 2.6 | 11.5 | — | 14.6 | 5.5 |
| ConZIC | 1.3 | 11.5 | — | 12.8 | 5.2 |
| **Ours** | | | | | |
| DenseNet161 | 4.20 | 12.5 | 30.1 | 17.0 | 6.6 |
| ResNet50 | 4.10 | 12.5 | 30.1 | 17.0 | 6.6 |
| ResNet50$_{v2}$ | 4.50 | 12.8 | 30.5 | 18.4 | 6.9 |
| WideResNet50$_{v2}$ | 4.20 | 12.7 | 30.3 | 17.7 | 6.9 |
| ResNet101v2 | 4.30 | 12.6 | 30.1 | 18.0 | 6.8 |
| ConvNeXt-Base$_{v2}$ | 4.40 | 12.8 | 30.2 | 18.6 | 7.1 |
| ResNet50$_{v2}$ | 4.50 | 12.8 | 30.5 | 18.4 | 6.9 |
| EfficientNetv2-S | 4.40 | 12.7 | 30.4 | 18.6 | 6.9 |
| ViT-B/16$_{pt}$ | 4.50 | 12.8 | 30.2 | 18.7 | **7.2** |
| ConvNeXtV2-B$_{pt}$@384 | 4.40 | 12.7 | 30.2 | 18.7 | **7.2** |
| BeiT-B/16 | 4.50 | 12.8 | 30.3 | **18.9** | 7.1 |
| DINOv2-Base | **4.60** | **13.0** | **30.7** | 18.7 | 7.1 |

Table 10: Zero-Shot Compositional Captioning Performance

While the results on CiDEr and SPICE are incremental compared to the results in Table 3, the n-gram metrics (B4, M and R-L) are boosted, which verifies our hypothesis that the low scores of B4 and M were attributed to the specific caption style and structure of the respective dataset (COCO).

## K.1 DOMAIN-SPECIFIC COMPOSITIONAL CAPTIONING

Since the LLM we used for compositional image captioning is generic and can adapt to any style and structure, compositional captioning is especially useful for generating captions tailored to specific domains. We explore alternative concept sets in Compositional Captioning. In the main manuscript, we reported results using the 20,000 most common English words as our concept set. Since the LLM remains fixed and functions as a composer, integrating detected concepts and verbs grounded in the image into a caption, we can seamlessly substitute the concept set with any domain-specific concept

| Method | B4 | M | R-L | C | S |
|---|---|---|---|---|---|
| ZeroCap | 2.6 | 11.5 | — | **14.6** | 5.5 |
| ConZIC | 1.3 | 11.5 | — | 12.8 | 5.2 |
| **Ours** | | | | | |
| MobileNetv3-L | 3.50 | 12.7 | 29.1 | 11.4 | 6.1 |
| ResNet50 | 3.60 | 12.7 | 29.3 | 12.0 | 6.0 |
| ResNet101v2 | 3.50 | 12.9 | 29.3 | 12.2 | 6.3 |
| WideResNet101v2 | 3.70 | 12.9 | 29.6 | 12.4 | 6.2 |
| ConvNeXt-Base | 3.80 | 12.8 | 29.5 | 12.7 | 6.2 |
| EfficientNetv2-S | 3.70 | 12.9 | 29.6 | 12.9 | 6.3 |
| ViT-B/16 (pt) | 3.80 | 13.1 | 29.5 | 13.2 | 6.5 |
| BeiT-L/16 | **3.90** | **13.2** | **29.6** | 13.4 | **6.6** |

Table 11: Composition Captioning Performance using the ImageNet-specific LF-CBM concept set

| Model | Top-1 | Orig. | $\Delta$ |
|---|---|---|---|
| ConvNeXt-Tiny | 82.19 | 82.52 | $-0.33$ |
| ViT-B/32 | 75.40 | 75.91 | $-0.51$ |
| Swin-Small | 82.63 | 83.20 | $-0.57$ |
| Swinv2-Tiny | 81.44 | 82.07 | $-0.63$ |
| CvT-21 | 80.45 | 81.27 | $-0.82$ |
| Swinv2-Small | 83.32 | 83.71 | $-0.39$ |
| ViT-B/16$_{pt}$ | 83.55 | 84.37 | $-0.82$ |

Table 12: Performance of our reformulated classifiers for additional models

set alternative. This allows for the generation of captions tailored to a specific domain. Here, we maintain the same set of verbs but explore the use of concepts specific to the ImageNet dataset. Since ImageNet lacks dedicated captions, we evaluate the domain-specific captioning by anticipating a decline in performance on the COCO captioning dataset. We use the ImageNet-specific concept set from (Oikarinen et al., 2023) and report zero-shot captioning performance in Table 11. As shown, we observe a decrease in all metrics. This shows that our method can readily produce captions for any domain. Finally, also note that we can control the style of the generations by simply prompting the LLM to compose the concepts and verbs in a specific style (e.g., humorous, positive, negative).

## L  PERFORMANCE ON ADDITIONAL MODELS

We report performance on additional models that were not included in the main manuscript in Table 12.

## M  PROCESS OF ZERO-SHOT IMAGE CAPTIONING

We remind readers of the mapping function, denoted as MLP, that transforms the visual features $f$ into the same space as textual features, producing $\tilde{f}$. A pre-trained language model $G$ is then optimized to generate a sentence that closely aligns with $\tilde{f}$. To preserve the generative power of $G$, we keep it frozen and apply prefix-tuning (Li & Liang, 2021), which prepends learnable tokens in the embedding space. We follow a test-time training approach to optimize learnable tokens for each test input on-the-fly. Our method builds upon the work of (Tewel et al., 2021).

A high-level overview of this process is illustrated in Figure 2. Using a pre-trained language model $G$, we prepend randomly initialized learnable tokens, referred to as prefixes, which guide $G$ to produce text that maximizes alignment with visual features. These learnable prefixes function as key-value pairs in each attention block, ensuring that every generated word can attend to them.

For each iteration $j$, at a timestep $ts$, we sample the top-$Q$ tokens from the output distribution of $G$, denoted as $G_{out}$, which serve as possible continuations for the sentence. These $Q$ candidate sentences are then encoded by a text encoder $T$, mapping them into the same embedding space as $\tilde{f}$. We compute the cosine similarity between each encoded sentence and $\tilde{f}$, resulting in $Q$ similarity scores. These scores are normalized with softmax and define a target distribution used to train $G_{out}$ via Cross-Entropy loss. The learnable prefixes are updated through backpropagation.

With the updated prefixes, $G$ is run again, and the most probable token is selected as the next word. This process is repeated for a predefined number of timesteps (up to the desired sentence length) or until the $<$ . $>$ token is generated. At the end of each iteration, a full sentence is generated. We conduct this process for 20 iterations, generating 20 sentences in total. The final output is chosen as the sentence with the highest similarity to the visual features $\tilde{f}$.

We also add the fluency loss from (Tewel et al., 2021) as well as other token processing operations. We refer readers to (Tewel et al., 2021) for more information. We use the small GPT-2 of 124M parameters as $G$. We also noticed that using a bigger $G$ (e.g., GPT-2 medium) does not enhance performance, indicating that a decoder with basic language generation knowledge is sufficient.

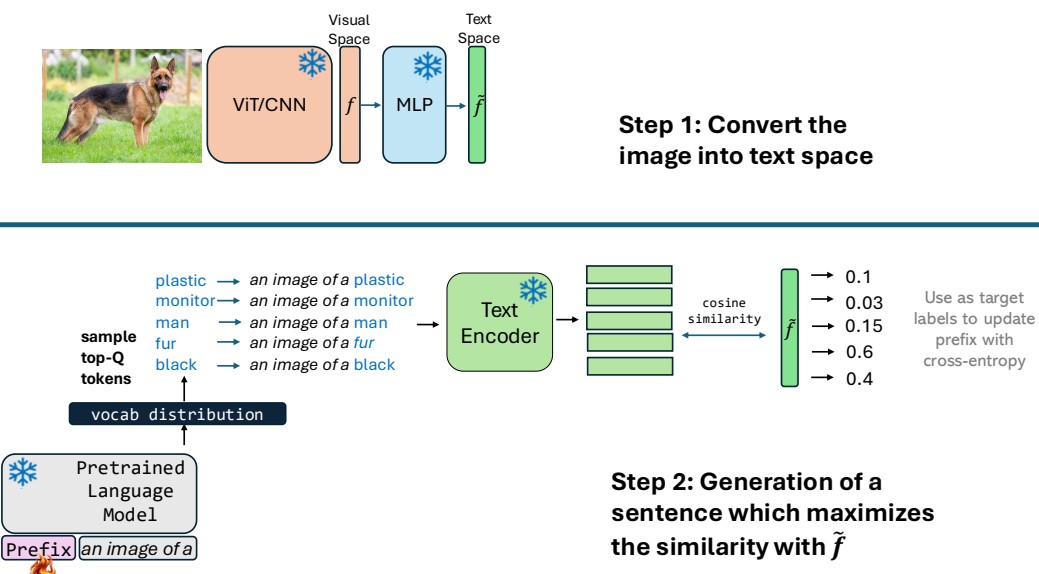

Figure 2: The process used to generate zero-shot captions using any pretrained language decoder (e.g., GPT-2). The process is shown for the first timestep ($ts = 1$) and first iteration ($j = 1$) with a hard prompt set as "an image of a". We apply prefix tuning while keeping the language decoder frozen, generating text that maximizes the similarity with the visual features.

## N  CONCEPT FILTERING

One of our concept filtering procedures requires us to know the terms corresponding to the parent and subparent classes (e.g., "fish" and "animal" for the class "tiger shark"), other species within the same category, and any synonyms of the target class name. In order to obtain this information, we used an LLM (gpt-4o-mini) with the following prompt:

```
Provide your answer to below as single-word comma separated.  If
you need to provide a term composed of 2 words, then separate each
into a single word.  For each class, provide its synonyms and
closely related names (e.g., other species of the category, its
superclasses such as bird, fish, dog, cat, animal...etc).  Here is
an example.  The ImageNet class is:  tench, tinca tinca.  Answer:
fish,animal,cyprinid,carp,vertebrate.
```

## O   USING ONLY CLASS NAMES

We remind readers from Section 3 that we only use the class names to formate the text prompt for the text encoder when training the MLP. In practice, we can go beyond class names by using resources like a class hierarchy from WordNet (Lin, 1998) (the original source where ImageNet was extracted from), or class descriptions extracted from a LLM as in CuPL (Pratt et al., 2023) or VCVD(Menon & Vondrick, 2023). However, this approach would be considered as "cheating. The original classifier implicitly learns the semantics, hierarchies, relationships and distinctive features of different classes. Explicitly providing additional information would not replicate the classifier faithfully since it would force the classifier to focus on predefined features or those we intend it to learn. Moreover, this would also leak information to downstream tasks such as CBMs and textual decoding of visual features, compromising the fairness of evaluation. For instance, if class descriptions were used in the training, the concepts in CBMs would align with those specified in the training prompts. For these reasons, we refrain from using any other additional information than the class names. We use the class names provided from `https://gist.github.com/yrevar/942d3a0ac09ec9e5eb3a`.

## P   GLOBAL CLASS-WISE QUALITATIVE EXAMPLES

In Figure 3, we present a probability distribution of global class-wise concepts. These are concepts detected for all images of a specific class, along with their frequency. We consider two semantically similar classes but distinctively different: "hammerhead shark" and a "tiger shark". We highlight in yellow the top concepts in "hammerhead shark" that are not present in "tiger shark". These concepts are "harpoon" and "lobster hammer", both which are distinctive to the head of the hammerhead shark and drive its prediction.

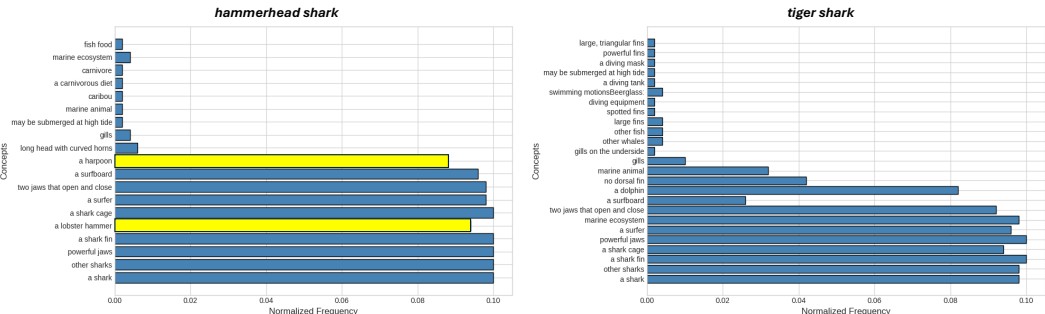

Figure 3: Global class-wise interpretability analysis with our Concept Bottleneck Model. We highlight in yellow the top concepts in "hammerhead shark" that are not present in "tiger shark", and therefore distinctive to "hammerhead shark".

## Q   WATERBIRDS/LANDBIRDS DATASET

There are two challenges associated with using the Waterbirds-100 dataset (Sagawa* et al., 2020) in our work. First, our method cannot transform a classifier trained on this dataset because learning meaningful text representations from merely two labels (waterbird and landbird) with our MLP is not feasible. Second, we cannot evaluate our ImageNet-trained models on this dataset due to its highly artificial nature, which results in numerous severe out-of-distribution samples for classifiers trained on ImageNet. Therefore, we manually curated our own waterbird/landbird dataset using ImageNet validation images. Specifically, for the waterbird images, we consider classes of birds from ImageNet that are at least 90% of the time found in water backgrounds— a statistic verified by manually inspecting 100 random training images of those birds. For those birds, we then select their images in the ImageNet validation set that appear on land backgrounds. We perform a similar procedure for the landbird class. For the landbird class, we encountered a problem. Although many ImageNet bird classes nearly always feature land backgrounds in their training images, we were unable to find a substantial number of validation images depicting these birds against water backgrounds (e.g., we could not find any image of the bird *robin* with water background in the ImageNet validation set). In

order to solve this issue, we utilized the Stable Diffusion 2.1 (Rombach et al., 2022) text-to-image generative model to generate images of those birds on water backgrounds. We ensured that the generated images of those birds have the correct physical and distinctive features of the bird. For all images, we always ensure that the background (water/land) is clearly visible. This leads us to a validation dataset of 140 images (70 images for each class). Specifically, for the waterbird class, all 70 images coming from the ImageNet validation set. For the landbirds class, 15 images come from the ImageNet validation set, and 55 are generated with Stable Diffusion.

## R    MULTI-CLASS CBM INTERVENTION CLASSES AND CONCEPTS

| Class | Concepts |
|---|---|
| tench | fish, freshwater, fins, dorsal, olive |
| english springer | dog, long ears, brown and white, playful, hunting |
| cassette player | portable, audio, tape, speakers, buttons |
| chainsaw | sharp, handheld, cutting, metal, wood |
| church | cross, tower, architecture, sacred, religious |
| french horn | curved, mouthpiece, musical instrument, orchestral, blow |
| garbage truck | large vehicle, wheels, clean, high load, lift |
| gas pump | fueling, hose, metallic, gasoline, handle |
| golf ball | small, white, round, rubber, dimples |
| parachute | fabric, fly, air, landing, strings |

Table 13: ImageNet classes and their five associated concepts we use in our multi-class CBM intervention experiment.

## S    USAGE OF LLMS FOR PAPER WRITING

We declare the usage of an LLM for the purpose of assisting in the writing process, specifically to aid in better expressing text, and polishing the overall style and readability of the text.

