# OpenReview forum: "CLIP-Free, Label-Free, Zero-Shot Concept Bottleneck Models"
_ICLR.cc/2026/Conference — ICLR 2026 Conference Withdrawn Submission_

### Official Review · Reviewer_tgqn · 2025-10-25

**Soundness:** 1
**Presentation:** 3
**Contribution:** 2
**Rating:** 2
**Confidence:** 4

**Summary:**

This paper addresses the construction of concept bottleneck models (CBMs), a type of interpretable neural network model, using existing pre-trained visual models. CBMs are classification models that take images as input and predict the final task label from the activations of intermediate concepts (interpretable text labels with finer granularity than the task label). While existing methods build CBMs based on pre-trained multi-modal models (e.g., CLIP) without additional annotation, this paper constructs CBMs using any existing pre-trained visual model. To achieve this, the paper first trains an MLP by matching the distribution of the original classifier output with a similarity score distribution between the MLP output and the text encoder output of the class label (TextUnlock). Next, this MLP is used to compute text label predictions for test images. The final class label is predicted by calculating the similarity between this prediction output and the concept label in the text feature space (Zero-shot CBMs). This approach eliminates the need for additional training in the subsequent CBM construction phase. The paper demonstrates that Zero-shot CBMs by TextUnlock can be constructed for many existing visual classification models, maintaining performance from the original classifier while providing explanations based on textual concepts.

**Strengths:**

- **S1.** The proposed method of adding MLP training to map the visual model's output features to the text feature space is straightforward and simple. **
- **S2.**  The paper conducts experiments across a wide range of visual models and confirms that task performance does not degrade.

**Weaknesses:**

- **W1.** The explanations provided by the proposed method offer limited interpretability regarding model behavior. First, in TextUnlock, the mapping between visual classifier output and the text feature space is trained on a limited vocabulary combination (e.g., ImageNet's 1000 classes), and it is not guaranteed that this aligns semantically with concepts in the text space. Furthermore, the concept-to-class calculation in Eq. (2) for Zero-shot CBMs inference relies on similarity in the text feature space, which does not guarantee that the classifier actually predicts the final label based on this concept. In other words, while existing CLIP-based CBMs guarantee the image-concept-label relationship through pre-training via large-scale text-image pairs, the proposed method constructs predictions a posteriori based on the image-label-concept-label relationship. Within this, the image-label inference remains a black box, and the label-concept-label inference is totally done in text feature spaces. This black-box nature is supported by the experimental results in Appendix E, which show that even when interventions remove important concepts, the accuracy of TextUnlock's zero-shot CBMs does not drop dramatically (i.e., not drop to zero). In my opinion, this limitation is critical because it prevents the model from directly providing explanations about its behavior to users.
- **W2.** The proposed method has limited directly applicable datasets. Experiments on zero-shot CBMs have primarily been conducted on ImageNet-1K. While Appendix G shows accuracy on Places, EuroSAT, and DTD, interpretability has not been evaluated. Furthermore, interestingly, synthetic data generated by Stable Diffusion is used for the Waterbirds (2-class) dataset used in Appendix E, as TextUnlock's MLP cannot be sufficiently trained on it, indicating that TextUnlock cannot be applied to such a simple classification. Currently, the condition remains unclear on which datasets the proposed method is applicable, and this limitation appears to be insufficiently discussed.
- **W3.** The paper directs readers to Appendix E in Section 4 for experiments concerning intervention, one of the most critical aspects in evaluating CBMs. This is unhelpful for readers and could even be considered a form of unfairness regarding page limitations. For many readers expecting a discussion on explainable models, Appendix E, which directly relates to the paper's research question, should be included in the main text.
- **W4.** The paper's claims and experimental content are inconsistent. While the paper's title and introduction suggest that constructing Zero-shot CBMs is its primary objective, Section 5 evaluates Zero-shot Image Captioning using TextUnlock. The paper should be revised to either make the construction of interpretable models its main claim or make TextUnlock its main claim.

**Questions:**

Please read the weaknesses and address the concerns.

---

### Official Review · Reviewer_tpWM · 2025-10-28

**Soundness:** 3
**Presentation:** 3
**Contribution:** 2
**Rating:** 6
**Confidence:** 3

**Summary:**

This paper proposes a method to convert any frozen visual classifier into a concept bottleneck model (CBM) while maintaining 3 characteristics: CLIP-free, label-free, and zero-shot. In particular, the label-free means they don't need annotations, CLIP-free means no access to CLIP models, and zero-shot since they derive a linear classifier at test time only. They do this by proposing to insert an MLP mapping visual encoder space into the shared text encoder space, to then do standard evaluation. The authors propose a KL divergence loss to train this MLP by distilling the visual encoder's linear classifier, and demonstrate how to do both concept discovery and concept-to-class prediction in a zero-shot manner. Finally, their empirical results show that they can outperform SOTA baselines on several tasks including classification and captioning.

**Strengths:**

* Experimental results are good.
* Simple and useful approach, no need to retrain the visual/text encoders.
* Comprehensive study over different model architectures.

**Weaknesses:**

* The use of different backbones across the proposed method and baseline methods makes it more difficult to quantify whether improvements in the experiment come from the proposed method or different backbone.
* Minor: some typos, e.g. Line 1190 (formate), Line 366 (Effectivenss), Line 131 (These, capitalized). Can run through a typo checker.

**Questions:**

* Line 215, "This systematic filtering guarantees ... free of terms that are overly similar...". Can you really guarantee that? The filtering process seems fairly adhoc.
* Appendix A, what is `text_feats = l2_norm(text_feats)` doing? Why is it still shape (N, text_dim)?
* In Table 2, I feel that some of the Ours models tested may not be directly comparable? For example, a baseline Supervised CBM has the ViT-L/14 model, but the only related Ours model is ViT-L/16_{v2} which is trained differently. Was there a reason why you didn't use the same model here?
* Out of curiosity, have you tried your approach on a CLIP baseline (i.e. using their visual/text encoders)? Does it improve the performance?
* In Table 1, are the results of the new formulation using the concept-to-class prediction, or just using MLP?
* What were the training times for your method?

---

### Official Review · Reviewer_9LsN · 2025-10-31

**Soundness:** 4
**Presentation:** 1
**Contribution:** 3
**Rating:** 4
**Confidence:** 4

**Summary:**

This paper presents TextUnlock, a zero-shot concept bottlenect models (CBMs) method without relying on a CLIP-based vision-language models. It alleviates existing limitations of previous CBMs by adapting a trained visual classifiers without requiring image-concept labels in a zero-shot manner. Specifically, given a language encoder, a MLP layer is trained on top of a visual classifier to map visual representations to the vision-language space by disilling the original classifier's predictions. Then, a zero-shot CBM approach is proposed, where the concept activation map is obtained by the similarity between the projected visual features and encoded text features from a concept back. Concept-to-Class procedure is done by calculating the similarity between the encoded features of class labels and concept bank. This simple approach can be applied to different choice of visual and language encoders. Experiments show that it does not harm the original classifier's performance while surpassing existing CBM mehods.

**Strengths:**

The proposed scheme for CBMs has some advantages. Despite the necessity of aligned visual and language representations, it does not requre the pre-trained CLIP models, which increasing the flexibility for the different choice of visual classifiers and text encoders. In addition, the approach is designed not to use the image-concept label, but rather distill the classifier's prediction it can preserve original classifier's performance. By this design, making CBMs is not limited to pre-trained CLIP models. Despite its simple methodology, experimental results show that this design is widely applicable to different combinations of visual classifiers and language encoders.

**Weaknesses:**

One major concern is regarding the claim and comparison with CLIP models made in the paper. From the CLIP models, as the visual and textual representations are already aligned, the training process of section 3.1 can be skipped and the process in section 3.2 can be directly applied to make CBMs. In terms of effectiveness, the performance comparison with CLIP models by directly taking them without any training process seems to be omitted. In addition, in terms of data efficiency, as the proposed method utilizes powerful text encoder trained on large-scale textual data to construct CBMs, the textual data scale is not considered when comparing data scales between CLIP and the proposed scheme. So, the claim on the data efficiency seems to be overclaim.

**Questions:**

Regarding the weaknesses stated above, a more rigorous comparisons to CLIP models in terms of effectiveness and data efficiency need to be presented. Specifically, the performance report of CLIP-oriented CBMs adapted by the process in section 3.2, in pair with the original CLIP model's zero-shot classification results is expected. Moreover, the overall claim regarding the data efficiency needs to be adjusted if the proposed scheme uses more data than CLIP, once the data scale used to train the text encoder is taken into account.

---

### Official Review · Reviewer_pQPj · 2025-10-31

**Soundness:** 3
**Presentation:** 3
**Contribution:** 3
**Rating:** 4
**Confidence:** 3

**Summary:**

While existing CBMs only handled CLIP models, this work proposes the TextUnlock technique, which can convert general models (not just CLIP) into CBMs. This requires training only a small MLP projector, and even this training is label-free, utilizing only class names. Consequently, the entire framework offers multiple advantages: it is clip-free, label-free, and requires no retraining of the main model. Applying the TextUnlock method to various models, including ResNet, ViT, ConvNeXt, and Swin Transformer, resulted in a maximum accuracy drop of 0.4%p on the ImageNet dataset, outperforming the CLIP backbone. This methodology also suggests the potential for extension to zero-shot image captioning. The Appendix also contains helpful content covering limitations, ablation studies on text encoder & MLP design, CBM intervention experiments, and results on other datasets.

**Strengths:**

- The work demonstrated that applying ‘TextUnlock’ results in a negligible average classification accuracy drop of 0.2%p across over 40 diverse models (Table 1). This overcomes the conventional tradeoff that “gaining explainability leads to performance degradation”.
- TextUnlock is successfully applied to over 40 diverse architectures, including ResNet, ViT, ConvNeXt, and Swin Transformer, demonstrating high generalizability not limited to specific architectures.
- TextUnlock successfully trained the projector MLP without explicit labels by using a train loss similar to Knowledge Distillation, not only for building CBM, but also for zero-shot image captioning on COCO images.
- Sufficient ablation studies were conducted throughout the main paper and appendix, along with analysis of limitations.

**Weaknesses:**

- The most significant difference between CLIP-based approaches and TextUnlock is likely to be observed when the dataset has fewer classes. Given TextUnlock's requirement to align image and text modalities using only class names, reducing the number of classes makes it difficult for the MLP mapping to be trained properly. Please provide the distribution of detected concepts from the Place365 and EuroSAT datasets discussed in the Appendix to facilitate discussion on this point.
- The primary conceptual weakness of TextUnlock is that it resembles ‘concept selection’ from a given list rather than ‘concept discovery’. While CBM's build does not require image-concept pair labels, it necessitates a predefined concept list like a ‘set of 20,000 textual concepts’ instead. It is evident that this list will have an absolute impact on both performance and explainability. This deviates from CBM's fundamental intent of understanding existing models and may lead to distorted results where models are forced to fit human intuition. Discussion is needed on how to view the fundamental purpose of CBM and concept detection.
- Continuing on the above shortcomings, although Appendix I conducted experiments showing this concept set generally works well, concerns remain that performance will decline and concept detection may fail when given OOD data.
- Compared to the emphasis on classification accuracy results in the paper, the perspective on concept detection and explainability seems relatively passive. Please present more explainability-related experimental results following prior studies like DN-CBM and DCBM.

**Questions:**

Personal Opinion – The work shows remarkable progress for extending CBM's scope from CLIP backbone to general models, without requiring additional labels or excessive training. However, separate from this technical advancement, considering the overarching direction of work aiming to understand which concepts a general model (or expert model) references during inference, the fact that concepts are selected from a predefined set seems to diminish the research's meaning. In the same point of view, emphasizing model explainability over minimal classification accuracy drop would be preferable. I would like to hear the authors' thoughts on this during the discussion session.

---

### Note · Authors · 2025-11-13

**Comment:**

We sincerely thank all reviewers for the time and effort invested in reading and evaluating our work. We value the feedback provided and appreciate the constructive comments that will help us further improve the manuscript. One of the reviewers (Reviewer tgqn) recommended a rejection (score 2). While we respect the reviewer’s perspective, we believe several aspects of the assessment may be based on misunderstandings of the material, and many of the concerns are already addressed in the supplementary material and referenced in the main manuscript. We also believe that other concerns are not grounds for rejection (e.g. simple organizational aspects of the paper or reporting additional tasks such as image captioning - which should actually strengthen the paper and not degrade it). The reviewer seems to simply find reasons to reject the paper.

We remain grateful for the constructive feedback from the other reviewers. However, based on our prior experience with the review process, a single rejection, with a soundness score of 1, significantly diminishes the likelihood of acceptance, even after revisions and rebuttal. Given this situation, we believe the most appropriate course of action is to withdraw the paper at this time.

**Withdrawal Confirmation:**

I have read and agree with the venue's withdrawal policy on behalf of myself and my co-authors.